# Poly I:C Pre-Treatment Induced the Anti-Viral Interferon Response in Airway Epithelial Cells

**DOI:** 10.3390/v15122328

**Published:** 2023-11-27

**Authors:** Hannah Mitländer, Zuqin Yang, Susanne Krammer, Janina C. Grund, Sabine Zirlik, Susetta Finotto

**Affiliations:** 1Department of Molecular Pneumology, Friedrich-Alexander-Universität (FAU) Erlangen-Nürnberg, Universitätsklinikum Erlangen, 91054 Erlangen, Germany; hannah.mitlaender@t-online.de (H.M.); zuqin.yang@uk-erlangen.de (Z.Y.); susanne.krammer@uk-erlangen.de (S.K.); janina.christine.grund@gmail.com (J.C.G.); 2Department of Medicine 1, Friedrich-Alexander-Universität (FAU) Erlangen-Nürnberg, Universitätsklinikum Erlangen, 91054 Erlangen, Germany; sabine.zirlik@uk-erlangen.de

**Keywords:** rhinovirus, asthma, epithelial cells, interferons, poly I:C

## Abstract

Type I and III interferons are among the most important antiviral mediators. Increased susceptibility to infections has been described as being associated with impaired interferon response in asthmatic patients. In this work, we focused on the modulation of interferon dysfunction after the rhinovirus infection of airway epithelial cells. Therefore, we tested polyinosinic:polycytidylic acid (poly I:C), a TLR3 agonist, as a possible preventive pre-treatment to improve this anti-viral response. In our human study on asthma, we found a deficiency in interferon levels in the nasal epithelial cells (NEC) from asthmatics at homeostatic level and after RV infection, which might contribute to frequent airway infection seen in asthmatic patients compared to healthy controls. Finally, pre-treatment with the immunomodulatory substance poly I:C before RV infection restored IFN responses in airway epithelial cells. Altogether, we consider poly I:C pre-treatment as a promising strategy for the induction of interferon response prior to viral infections. These results might help to improve current therapeutic strategies for allergic asthma exacerbations.

## 1. Introduction

Asthma is a respiratory disease associated with chronic airway inflammation and changes in the lung tissue architecture resulting in the worsening of the lung function with airflow obstruction during asthma exacerbations. The latter events are often triggered by allergens and respiratory viruses and especially the Human Rhinovirus (RV), the main pathogen causing the common cold. After entering the host body, these viruses first target airway epithelial cells in which they replicate. The RV is a non-enveloped, single-stranded positive-sense RNA virus of the Picornavirus family. The 160 serotypes have been classified into the three species: RV-A, RV-B, and RV-C [1,2]. Depending on the serotype, the RV strains use different receptors located on the cell surface for their uptake into the cells. Within the cell, the viral genome is recognized by pattern recognition receptors in the endosome (Toll-like-receptors (TLR) 3, 7 and 8) and in the cytosol (by retinoic acid inducible gene I (RIG-1) and melanoma differentiation-associated protein 5 (MDA-5)) [3]. Polyinosinic:polycytidylic acid (poly I:C) is a synthetic immunostimulant consisting of dsRNA that can interact with TLR3 expressed on the membrane of cellular endosomes and is able to activate the downstream signaling of TLR3. This signaling cascade leads to the translocation of nuclear factor ‘kappa-light-chain-enhancer’ of activated B-cells (NF-κB) and interferon regulatory factors (IRFs) 3 and 7 to the nucleus, causing the transcription of pro-inflammatory cytokines, chemokines, and interferon (IFN) type I and III [4,5,6,7]. The released IFNs type I and III then bind to their IFN receptors in an autocrine and paracrine manner. The induced downstream signaling leads to an increased expression of interferon-stimulated genes (ISGs); these include antiviral effector molecules such as 2’-5’-oligoadenylate synthetase 1 (OAS-1) [8,9,10]. Their release induces an antiviral status in the cells, preventing the spread of viruses to other uninfected cells. In return, the respiratory viruses must evade those immune responses, to remain in the tissue. Control and asthmatic patients show great differences also on a cellular level when it comes to the barrier function and the defense against external pathogens. Several studies have implicated that the interferon immune responses to RV in asthmatic patients are deficient or delayed compared to a control group [3,11,12]. However, this topic is controversially discussed and needs further research. Therefore, in our human cohort study “AZCRA”, we recruited healthy controls and asthmatic patients from which we obtained nasal epithelial cells, lung function measurements, and clinical data. In this study, we focused on the modulation of interferon dysfunction seen in the airways of asthmatic subjects after rhinovirus (RV) infection. Thus, we investigated the impact of in vitro rhinovirus infection on interferon responses in airway epithelial cells and looked for therapeutic strategies to overcome their anti-viral deficiency in asthma. We found decreased interferon levels after RV infection and especially in the asthmatic patients, while the RV load was increased. Additionally, in an airway epithelial cell line, we investigated poly I:C and its effect on the immune response to RV infection and found that pretreatment with the TLR3-agonist caused a great increase in IFN I, II, and III and in effector protein levels. Therefore, poly I:C could play a role in restoring the interferon response to reduce asthmatics’ susceptibility to infections and therefore in future therapeutic strategies as well.

## 2. Materials and Methods

### 2.1. Human Cohort Study “AZCRA”

The human cohort study “AZCRA” (Investigation of the role of cytokines, chemokines, and their receptors in the inflammatory process in asthma patients) is conducted at the University Hospital Erlangen and was approved by the ethics review board of the Friedrich-Alexander-University of Erlangen-Nuremberg (315_20B). The study was registered in the German Clinical Trial Register (DRKS00023843). Asthmatic patients and non-atopic, non-asthmatic control patients between 18 and 65 years of age were recruited, and informed written consent was obtained. At the Baseline Visit B0, during a non-symptomatic phase, blood and nasal swabs were taken, lung function was measured, and a questionnaire about clinical data was collected. General clinical characteristics describing the control and asthma cohorts and the differences between the two groups are reported in Table 1, and the single data sets for all the patients can be found in Appendix A.

All our patients during the baseline visit were asymptomatic and specifically showed no respiratory symptoms. In our asthma group, all patients were previously classified as an allergic or mixed phenotype, meaning that have allergic asthma in combination with another phenotype that causes asthma symptoms. All of the asthma patients but one had other atopic diseases, for example, clinically relevant allergies, atopic rhinitis, or atopic dermatitis. Most of our patients (77.26%) were currently on asthma medication—mostly a combination of inhaled glucocorticoids and long-acting beta-agonist. More than half of our asthma patients had a family history of asthma or other atopic diseases (54.55%) compared to only 21.05% in our control group.

### 2.2. Nasal Epithelial Cell Culture

Nasal swabs of the study participants were obtained and transported in UTM tubes (Copan, Brescia, Italy; all reagents listed in Appendix A). The cells were collected from the swabs by centrifugation (450× *g*, 5 min, 4 °C) and were treated with DNAse I (1.5 mg/mL, diluted in H_2_O; Sigma-Aldrich, St. Louis, MO, USA) for 20 min and washed with medium afterwards. Then, depending on the condition, the nasal epithelial cells (NECs) were freshly infected with RV-A1b (protocol below) or directly seeded into collagen-coated 48 well plates in 110 µL Pneumocult+ medium (PneumoCult-Ex-Plus Medium (Stemcell™, Grenoble, France) + 1% Antibiotic-Antimycotic (Gibco™, Waltham, MA, USA) and 0.5% Gentamicin (10 mg/mL; Sigma-Aldrich, St. Louis, MO, USA)), and 90 µL Pneumocult++ medium (Pneumocult+ medium +5% FBS (Sigma-Aldrich, St. Louis, MO, USA) +2% Sodium Bicarbonat (Gibco^TM^, Waltham, MA, USA)). After 72 h, the cells were harvested and used for FACS analysis, or RNA was extracted for qPCR analysis and RV detection.

### 2.3. A549 Cell Culture

The human A549 (ATCC^®^ CCL-185™) lung epithelial adenocarcinoma cell line was purchased and authenticated from the ATCC bank (Manassas, VA, USA). For the experiments, the cells were seeded in 5 × 10^5^ in 6-well plates (Greiner Bio-one, Kremsmünster, Austria) with 2 mL medium (See Appendix A) containing 100 mg/dL glucose (1:1 mix of Gibco™ RPMI 1640 mediums (Thermo Fisher Scientific, Waltham, MA, USA) containing either 0 mg/dL or 200 mg/dL glucose, supplemented with 10% heat-inactivated FCS (Sigma-Aldrich, St. Louis, MO, USA), 1% Penicillin–Streptomycin (Anprotec, Bruckberg, Germany), or 1% 2 mM L-Glutamine (Anprotec, Bruckberg, Germany). The cells were infected with RV-A1b as described below and/or stimulated with poly I:C (working concentration: 20 μg/mL, Sigma-Aldrich, St. Louis, MO, USA) at different time points relative to the RV-A1b infection (see experimental design). Cells were incubated at 37 °C, 5% CO_2_ and 96% humidity. They were cultured for 48 h for RNA analysis and supernatant, and 72 h for supernatant and flow cytometry analysis. Cells were detached using Trypsin-EDTA (Anprotec, Bruckberg, Germany) and then counted using Trypan Blue solution (Sigma-Aldrich, St. Louis, MO, USA) in a Neubauer chamber in accordance with the formula ((Q1 + Q2)/2) × 2 × 10,000 × x mL. 

### 2.4. RV Infection

In our study, we utilized the RV-A1b serotype, which is classified as part of the RV-A species and uses the low-density lipoprotein receptor (LDLR) for its uptake into the cells [13]. The virus propagation in HeLa cells was previously described [14]. For the infection, 0.5 mL RV-A1b (TCID50) solution was added for 1 Mio. Cells, then resuspended and incubated for 1 h at 33 °C in a laboratory shaker (Edmund Bühler GmbH, Bodelshausen, Germany). Then, the cells were washed and resuspended in medium and seeded back into the well plates. 

The infection was detected by Rhinovirus qPCR. It was performed using iTaq™ Universal SYBR Green Supermix (Bio-Rad Laboratories, Hercules, CA, USA), using primers from Eurofins-MWG-Operon (Ebersberg, Germany; RV). The qPCR was performed on a CFX-96 Real-Time PCR Detection System (BIO-RAD, Munich, Germany), and was analyzed using CFX Manager Software (Version 3.0). The RV-A1b forward primer 5′-CCA TCG CTC ACT ATT CAG CAC-3′ and the reverse primer 5′-TCT ATC CCG AAC ACA CTG TCC-3′ were used for qPCR. The relative levels of transcripts were calculated as 2^−ΔΔCT^, with HPRT used as the housekeeping gene and an internal control. 

### 2.5. qPCR

After harvesting, the cells were stored in Qiazol Lysis^®^ Reagent (Qiagen, Hilden, Germany) for RNA extraction according to the manufacturer’s instructions. The extracted RNA was converted into cDNA with the RevertAid™ First Strand cDNA Synthesis Kit (Thermo Scientific™, Waltham, MA, USA) using the manufacturer’s protocol. The quantitative qPCR analyses were performed with iTaq™ Universal SYBR Green Supermix (Bio-Rad Laboratories, Hercules, CA, USA) and using the gene specific primers bought from Eurofins-MWG-Operon (Ebersberg, Germany) and listed in Appendix A. The qPCR was performed using the CFX-96 Real-Time PCR Detection System (BIO-RAD, Munich, Germany), and was analyzed by the CFX Manager Software (Version 3.0). The relative levels of transcripts were calculated as 2^−ΔΔCT^, with HPRT as an internal control and relative to the CT of the control group. Each experiment was performed in duplicates, and values higher than 35 cycles were excluded from the analysis.

### 2.6. ELISA for IFN-λ

IFN-λ concentrations in cell culture supernatants were measured using the enzyme-linked immunosorbent assay (ELISA) technique according to the manufacturer’s instructions (human IL-29/IL-28B (IFN-lambda 1/3) ELISA (DuoSet Elisa Development System); R&D Systems, Minneapolis, MN, USA). 

### 2.7. Flow Cytometry 

Nasal epithelial cells were analyzed for their surface antigen expression by flow cytometry analysis. They were washed twice with FACS buffer (PBS (Anprotec, Bruckberg, Germany) +2% FCS (Sigma-Aldrich, St. Louis, MO, USA)), and then Fc-Block (1:100, BD Pharmingen, Heidelberg, Germany) was applied for 5 min at 4 °C in the dark to prevent non-specific antibody binding. Then, surface staining with anti-EpCAM antibodies (APC, Biolegend, San Diego, CA, USA) to identify epithelial cells was performed for 30 min in the dark at 4 °C temperature. The detection of the stained cells was performed using a BD FACS Canto II flow cytometer (BD Biosciences, Heidelberg, Germany). The acquired FCS files were then analyzed using Flow-Jo v10.2 (FlowJo, LLC, Ashland, OR, USA) software. 

### 2.8. Statistical Analysis

The statistical analyses were performed with Prism version 8 for Windows (GraphPad, San Diego, CA, USA). The differences were evaluated for significances (*: *p* ≤ 0.05; **: *p* ≤ 0.01; ***: *p* ≤ 0.001; ****: *p* ≤ 0.0001) with the Kruskal–Wallis test for comparisons of multiple non-parametric groups and the One-Way ANOVA for multiple parametric groups. Mann–Whitney U-test and unpaired *t*-test were used to assess significances between two non-parametric and parametric groups, respectively. The correlation of non-parametric data was performed with Spearman correlation and simple linear regression, and the parametric data analysis was performed with Pearson correlation. Graphs show the mean values ± sem and were created using GraphPad Prism 8, Windows.

## 3. Results

### 3.1. Asthmatic Patients Had Increased Airway Infections in the Last 12 Months and Decreased Levels of IFN Type I and III in Nasal Epithelial Cells (NECs) Compared to Control Subjects

In the adult human cohort study “AZCRA” (Investigation of the role of cytokines, chemokines, and their receptors in the inflammatory process in asthma patients), we recruited healthy, non-asthmatic and non-atopic, controls and asthmatic adults. Both groups attended a baseline visit, B0, during which clinical data and nasal swabs were collected (Figure 1a). Comparing the number of airway infections in the last 12 months in both groups, we found no significant difference between the two groups. However, several asthmatic participants had up to six (6) airway infections in the last 12 months, compared to the maximum of two (2) respiratory infections in the control group (Figure 1b). Comparing the lung function measurements between the two groups, we found, by trend, worse results in the asthma group. Next, we wanted to analyze the antiviral immune response of the airway epithelium to RV and thus collected nasal swabs. We infected the cells freshly after collection with RV or used them as control without infection (CN) and cultured them for 3 days. On day 0 and 3, we performed FACS staining on some of our samples from control patients to ensure that the cultured cells were nasal epithelial cells (NECs). We checked the expression of the epithelial cell marker EpCAM on the cells, which stands for epithelial cell adhesion molecule, which is expressed only on the epithelium. This marker belongs to the type I transmembrane glycoproteins. It is involved in the regulation of cell adhesion, proliferation, migration, and stemness. Here, we found that there was a proliferation of EpCAM-expressing cells between day 0 and day 3 (Figure 1c–e). After 72 h, we first assessed the viral load by RV-A1b qPCR in all our infected conditions and found higher levels in NECs derived from asthmatic patients (Figure 1f). Then, we measured the IFN-β and IFN-λ mRNA in the uninfected and RV-infected condition and found higher levels by trend in the control group and lower levels in the RV-infected NEC culture derived from subjects with asthma (Figure 1g,h), indicating a defect in interferon response in nasal epithelial cells derived from asthmatic subjects and especially upon RV-A1b infection in combination with higher RV-A1b load in the cells. 

### 3.2. TLR3 Correlated Positively with RV-A1b Clearance in Epithelial Cells

We next wanted to investigate the possibility to restore anti-viral interferon responses in the airways in asthma and thus analyzed immunomodulatory anti-viral substances in the epithelial cell line A549. Toll-like receptor 3 (TLR3), which recognizes double-stranded RNA (dsRNA), has a central role in the host response to viruses. We thus thought of poly I:C as a good candidate due to its binding to TLR3 (Figure 2a). Thus, we correlated the RV-A1b mRNA expression and TLR3 mRNA upon RV-A1b infection in A549. Here, we found an inverse correlation between RV-A1b mRNA and TLR3 mRNA expression in infected A549 cells (Figure 2b). These data indicated to us the involvement of TLR3 on RV-A1b clearance. Next, we investigated the effect of poly I:C pre-treatment on TLR3 mRNA expression (Figure 2c). By trend, we found an induction of TLR3 when poly I:C was applied 12 h before RV-A1b infection. These results indicated that pre-treatment with Poly I:C 12 h before RV-A1b infection might be a good strategy to induce IFN responses in airway epithelial cells.

### 3.3. Pre-Treatment with Poly I:C before RV-A1b Infection Induced IFN-λ and OAS-1 Antiviral Immune Responses in Lung Epithelial Cells

We next hypothesized that the time-point of the poly I:C treatment could shape the interferon response to RV-A1b infection. We then tested two time points of administration of poly I:C: one time-point 12 h before and one time-point during the viral infection (Figure 3a). Here, we found an induced antiviral effect of the poly I:C pre-treatment (12 h before RV-A1b infection) on OAS-1, IFN-λ, and IFN-β expression (Figure 3b–d). For IFN-α mRNA levels and the expression of its receptor, we found a significant induction by poly I:C treatment alone and also for poly I:C administered 12 h before the RV-A1b infection by trend (Figure 3e,f). RV-A1b load in the cells was measured by qPCR and was slightly lower after pretreatment with poly I:C compared to the cells infected without poly I:C or when simultaneously administered during infection. These results indicate an immunomodulatory, antiviral effect of the poly I:C pre-treatment, which might enable the immune system to induce interferons to fight the subsequent infection more efficiently.

## 4. Discussion

Respiratory viruses primarily infect airway epithelial cells, which are the first defense barrier of the airway mucosa [15,16,17,18,19]. The host responds to viral infections by releasing antiviral factors like interferon type I (interferon α), II (interferon γ), and III (interferon λ) [20,21,22,23], leading to cell death of infected cells. As a result, during repeated infections there is a disruption of the epithelial barrier with increased risk of virus dissemination to other distant sites through the sub-epithelial tissue to encounter the peripheral blood immune cells [24,25,26,27], this happens especially in the epithelium of asthmatic patients, since the epithelial integrity is already impaired [28,29,30]. 

In our human cohort study “AZCRA”, the asthma group has had tendentially more viral infections in the last 12 month than the control group and showed worse lung function measurements. Additionally, in nasal epithelial cells of asthmatic patients, we found a lower expression of IFN-β and IFN-λ mRNA. This might suggest reduced or dysregulated antiviral immune response via decreased interferon production in the upper airways in asthma patients. This is supported by the increased RV-A1b load in the asthma group. 

Our data showed that the interferon levels were decreased, especially after RV-A1b infection. This seems contradictory since in the literature, it was reported that RV-A1b induces interferon responses, with the peak at around 8–48 h and decreasing levels after that [31,32]. At the 72 h timepoint that we chose, we found decreased IFN levels in the nasal epithelial cells after RV-A1b infection. This might be the result of the negative feedback loop known for IFN, which might be active at 72 h post infection [33]. 

We then looked for a way to rescue this impairment of the interferon-mediated immune response and investigated the influence of the TLR-3 agonist poly I:C. To do so, we used it to mimic contact with the viral RNA to prime the cells for contact with the real RV genome during infection. We found that TLR3 mRNA expression correlated with higher RV-A1b clearance. Furthermore, we found an effective induction of IFN type I and III production and the effector antiviral molecule OAS-1, when the cells are pretreated with poly I:C 12 h before the infection. After binding to double-stranded viral RNA, OAS-1 produces 2′-5′ linked oligoadenylate (2’5 A), which then activates ribonuclease L (RNase L) resulting in viral RNA degradation and thus the inhibition of virus replication [18]. Interestingly, this induction only occurred when the stimulation of poly I:C and RV-A1b was consecutive and not during simulant stimulation. In the pretreated cells we additionally found slightly lower RV-A1b load by qPCR. This indicates that poly I:C, in fact, might be able to support the cells to remove the virus from the cells. 

The goal for our research was to investigate the effect of poly I:C on RV-A1b infection in an in vitro cell line model. However, it is necessary to eventually translate these results into clinical practice. That is why the question arises if poly I:C could be used in a medical setting. Human in vivo studies investigating poly I:C have been conducted using its derivate Poly-ICLC (Hiltonol, Drug bank number: DB17293), which is stabilized with poly-L-lysin and carboxymethyl cellulose to resist glycolysis in serum [34]. Studies showed that poly-ICLC showed only mild side effects and was well tolerated by the participants; in patients with HIV (human immunodeficiency virus), it increased the innate immune responses temporarily [35,36]. While the first trials and their results look promising, more studies investigating the use of poly-ICLC in humans must be conducted to ensure safety and effectiveness.

Another obstacle occurs for the use of poly I:C in humans. TLR-3 is known to have contradictory properties in allergic diseases as well as in viral infections. On one hand, its activation induces the production of interferons, which is needed to contain the viral infection, on the other hand it can have detrimental effects and induce inflammation. It leads to the upregulation of FcεR1 expression on activated dendritic cells and thus to the recruitment of Th2 cells, which leads to excessive inflammation [37,38]. Murine experiments showed induced exacerbations by poly I:C application in an established asthma model [39]. Therefore, we thought about ways to activate only the antiviral effects released by the epithelial barrier. In a mouse model, the dendritic cell (DC)-specific administration of poly I:C in combination with DC-targeted protein was successful in improving the efficiency of vaccination-induced T-cell immunity [40], which opens possibilities to induce poly I:C responses only in designated cell types. Thus, translating into our studies, an epithelial-cell-specific delivery of poly I:C, or its derivate Poly-ICLC, could help to induce defense against infections, without stimulating Th2 immunity and a pro-inflammatory response. Then, it could be used as a pre-treatment strategy before RV infections in order to prevent asthma exacerbations.

Next steps should include identifying a delivery agent that works similarly to the one described for dendritic cells. This agent would bind specifically to epithelial cells and not to immune cells and thereby reduce the damaging effects of poly I:C on the lung tissue and only improve the immune defense against the viral infection. Therefore, co-culture experiments should be carried out with epithelial cells and various immune cells to test if this approach is promising.

To further confirm our results, it would be necessary to investigate the interferon secretion and its response to poly I:C in other human epithelial cell lines and in primary airway epithelial cells derived from asthmatic and control patients. Furthermore, other antiviral immune responses, such as proinflammatory interleukins, must be taken into consideration. Taken together, our study suggests that the utilization of poly I:C as a pre-treatment holds promise in potentially enhancing therapeutic approaches for managing asthma exacerbations. This is achieved by triggering Interferon responses in epithelial cells prior to RV infection often taking place during common cold season. 

## Figures and Tables

**Figure 1 viruses-15-02328-f001:**
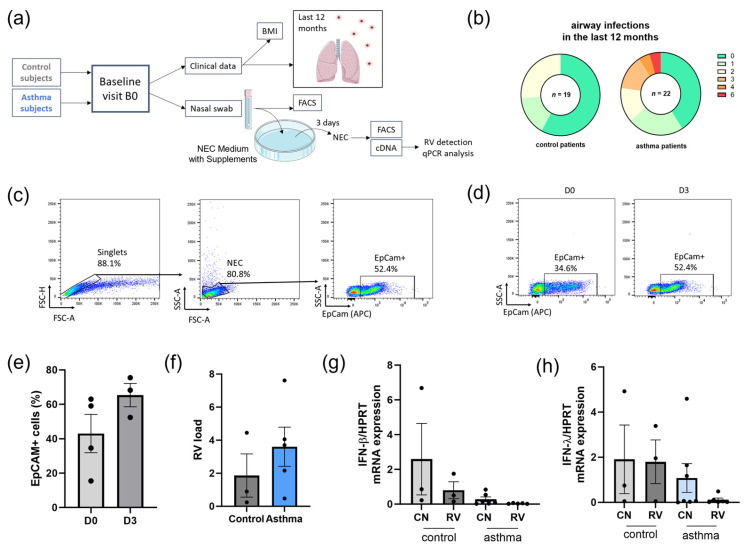
Asthmatic patients had more frequent airway infections and deficient interferon responses in NECs. (**a**) Experimental design of human cohort study “AZCRA”, NEC= Nasal epithelial cells. Partly generated with Servier Medical Art, provided by Servier, licensed under a Creative Commons Attribution 3.0 unported license. (**b**) Number of airway infections in the last 12 months. Left: control patients (*n* = 19). Right: asthma patients (*n* = 22). (**c**–**e**) Flow cytometry analysis of EpCam+ NECs (%) from control patients directly from nasal swab (D0) or after cell culture for 3 days (D3), and a representative dot plot was shown for each group. Color gradient shows density of cells in an area (red = high density, blue = low density). (**f**) RV-A1b load by qPCR in RV-infected NEC from control and asthma patients (*n* = 3/5). (**g**,**h**) mRNA level of IFN-β (*n* = 3/3/6/7) and IFN-λ (*n* = 3/3/7/6) in NECs quantified by qPCR, normalized to HPRT. Unpaired t test and ordinary one-way ANOVA were used for statistical analysis. Graph shows mean ± sem. Color legend: Grey: control patients, blue: asthma patients.

**Figure 2 viruses-15-02328-f002:**
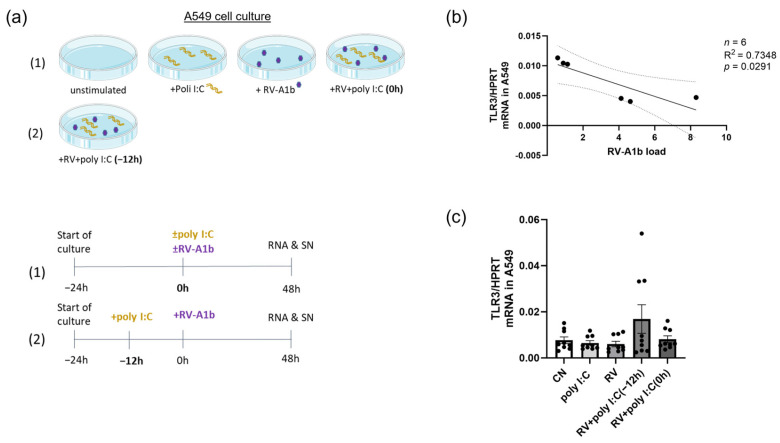
TLR3 correlated positively with RV-A1b clearance from epithelial cells. (**a**) Experimental design of the study: Culture of uninfected and RV-infected A549 cells in medium with poly I:C administered either 12 h before the infection or during the infection with RV-A1b. Below: timetable of the culture. Partly generated using Servier Medical Art, provided by Servier, licensed under a Creative Commons Attribution 3.0 unported license. (**b**) Correlation of TLR3 mRNA level and RV-A1b mRNA expression in RV-infected A549 as quantified by qPCR (*n* = 6). (**c**) Pre-treatment with Poly I:C induced TLR3 mRNA expression in A549 48 h after RV-A1b infection, analyzed by qPCR, normalized to HPRT (*n* = 9). Pearson correlation coefficient was used for the linear correlation analysis and ordinary one-way ANOVA was used for statistical analysis. Graphs show mean ± sem.

**Figure 3 viruses-15-02328-f003:**
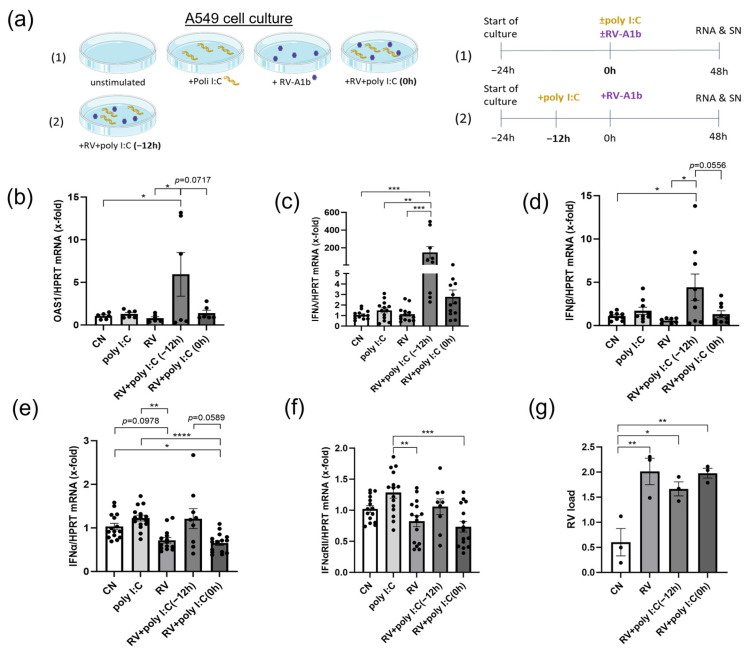
Pre-treatment with poly I:C before RV-A1b infection induced IFN-λ and OAS-1 antiviral immune responses. (**a**) left: experimental design: culture of uninfected and RV-A1b-infected A549 cells, with poly I:C administered either 12 h before the infection or during the infection with RV-A1b. right: timetable of the culture. Partly generated with Servier Medical Art, provided by Servier, licensed under a Creative Commons Attribution 3.0 unported license. (**b**) mRNA levels of OAS1 (*n* = 6), (**c**) IFN-λ (*n* = 13/13/13/9/12), (**d**) IFN-β (*n* = 9/9/7/9/9), (**e**) IFN-α (*n* = 15/15/14/9/15) and (**f**) IFNαRII (*n* = 15/15/15/9/15) in A549 after 48 h were analyzed by qPCR, normalized on HPRT. (**g**) RV-A1b load assessed by qPCR, normalized to HPRT (*n* = 6). Significance between groups were detected by Ordinary one-way ANOVA or Kruskal–Wallis test. Graphs show mean ± sem. *: *p* ≤ 0.05; **: *p* ≤ 0.01; ***: *p* ≤ 0.001; ****: *p* ≤ 0.0001.

**Table 1 viruses-15-02328-t001:** Control und asthma cohorts of the “AZCRA” study.

Clinical Characteristics	Control	Asthma	*p*-Values
Number of subjects	19	22	
Male gender	10 (52.63%)	10 (45.5%)	
Female gender	9 (47.36%)	12 (54.5%)	
Age (in years)	39 ± 15.83 (21–64)	42.18 ± 13.09 (23–63)	
BMI (in kg/m^2^)	23.52 ± 4.220 (17.15–32.50)	26.75 ± 4.973 (21.80–44.79)	0.0214
Number of airway infections in the last 12 months	0.6842 ± 0.8852 (0.0–2.0)	1.364 ± 1.620 (0.0–6.0)	0.1886
Allergic asthma phenotype	/	22 (100%)	
Allergic rhinitis	0 (0%)	17 (77.26%)	
Asthma medication	0 (0%)	17 (77.26%)	
Family history of atopy/asthma	4 (21.05%)	12 (54.55%)	0.0529
Lung function FEV1 (%)	100.6 ± 10.92 (84.0–126.0)	93.91 ± 13.41 (74.0–126.0)	0.0919
Lung function FVC (%)	99.68 ± 10.47 (80.0–117.0)	96.32 ± 10.73 (81.0–120.0)	0.3173
Lung function: FEV1/FVC (%)	82.50 ± 7.59 (69.38–97.24)	78.55 ± 7.984 (64.24–94.54)	0.1142

BMI: Body mass index. FEV1: One-second capacity. FVC: Forced vital capacity. Data are presented as n (%) or mean ± Standard Deviation (minimum–maximum).

## Data Availability

The datasets generated during the current study are available from the corresponding author on request.

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
