# Peer review of "Poly I:C Pre-Treatment Induced the Anti-Viral Interferon Response in Airway Epithelial Cells"

_viruses, 2023, doi:10.3390/v15122328_

Round 1

Reviewer 1 Report

Comments and Suggestions for Authors

Thank you very much for the invitation to review this interesting and confirmatory piece of work.

The manuscript is very clearly written. 

I have some minor comments. Please see below:

Minor comment 1

Could you please explain why you used the 12-hour timepoint for pre-treatment with poly: IC before RV infection?

Minor comment 2

I can see that the abbreviations are spelled out inside the manuscript. However, in Figure 1 it is difficult to understand the experimental conditions at the box plots. Could you please add abbreviations to the figures as well?

Minor comment 3

Are there any available drugs that target poly: IC that can be repurposed to be used in asthma? You could make a table from DrugBank online.

Author Response

Reviewer 1

Thank you very much for the invitation to review this interesting and confirmatory piece of work.

The manuscript is very clearly written. 

Answer

The authors thank the reviewer for evaluating their work.

Reviewer

I have some minor comments. Please see below:

Minor comment 1

Could you please explain why you used the 12-hour timepoint for pre-treatment with poly: IC before RV infection?

Answer

 We reasoned that time point would be therapeutic important, based on the dynamic of IFN production by the cells. In the literature we found that the peak of IFN production after viral infection is around 8-48 hours. We chose 12 hours as pretreatment with an agent that mimics viral RNA to optimize responses of the host target cells before the rhinovirus is added. In future studies it would also be helpful to conduct experiments with more time-points to find the optimal time-point of medication administration.

Minor comment 2

I can see that the abbreviations are spelled out inside the manuscript. However, in Figure 1 it is difficult to understand the experimental conditions at the box plots. Could you please add abbreviations to the figures as well?

Answer: We changed the figure consistently and hope that it is clearer now.

Minor comment 3

Are there any available drugs that target poly: IC that can be repurposed to be used in asthma? You could make a table from DrugBank online.

Answer: Thank you for the helpful comment. There are human in vivo studies investigating poly I:C‘s derivate Poly-ICLC, which is used since it is more stable(1). The use of poly-ICLC is still investigational, but promising: In patients with glioblastoma it was used to boost immunotherapeutic approaches (2) and in HIV patients it was shown that Poly-ICLC induces innate immune responses in PBMCs opening the possibilities to use it as an adjuvant for therapies (3). Poly-ICLC is registered on DrugBank as Hiltonol (DB17293).

Reviewer 2 Report

Comments and Suggestions for Authors

Summary

In this study, the authors recruited healthy and asthmatic adults from the human cohort study AZCRA. They found that the asthmatic subjects experienced up to six respiratory infections in the previous 12 months compared to a maximum of two infections for the healthy group. Nasal swabs were collected and cultured in the presence or absence of RV1B for 72 hours. They found that the cultures from asthmatic subjects had deficient IFNb and IFNL production especially after RV infection. The authors then tested the hypothesis that pretreatment with Poly-IC could restore IFN production and provide protection against infection. They found that pretreatment with Poly-IC 12 hours prior to RV infection enhanced TLR3 expression in A549 cells and antiviral immune responses. The authors conclude that poly I:C pre-treatment is a promising candidate therapeutic for the prevention of allergic asthma exacerbations.

Major comments

1. Study design. The study design is described as a human cohort study however it was also registered as a clinical trial. The authors should clarify if this study is an observational study or a clinical trial. If the study is a clinical trial, they should provide details of the intervention.

2. Study population. The study population is described in Table 1. The table currently lacks some clinical variables that are important to describe the asthma patients. Some additional variables that could be included are:

Family history of atopy

Family history of asthma

Allergen-specific IgE levels

Total IgE levels

SPT to panel of aeroallergens

Lung function (spirometry)

Current medications

3. Collection of nasal swabs. It is important to understand if the nasal swabs were collected when the patients were asymptomatic or not at the time of sampling. It is also important to understand if the patients were currently on medications at the time of sampling.  

4. The authors measured the purity of the nasal epithelial cell cultures, but this was not discussed in the manuscript. The authors should present the purity data for asthmatic and healthy subjects before and after culturing and report on any differences between the groups. 

5. In figure 1e and 1f, IFN production in the RV stimulated cultures was lower than in the control cultures. This result was unexpected and is not consistent with the literature. The authors should check if the house keeping gene is stable and appropriate for this experimental system.

6. RV loads. The authors should measure RV loads using qPCR throughout the study to determine if this is higher in nasal epithelial cell cultures from subjects with asthma and additionally if the viral loads are altered by Poly-IC pretreatment.

7. In the discussion, the authors should comment on the utility of Poly-IC in clinical settings and perhaps discuss other options such as inhaled IFNb or Poly-ICLC.

8. The authors conclude that poly I:C pre-treatment is a promising candidate therapeutic for the prevention of allergic asthma exacerbations. However, this conclusion is not supported by the data given that the atopic status of the asthma patients was not confirmed and only 12/22 of the asthmatic subjects have allergic rhinitis.

9. The authors could consider stratifying the subjects into four groups based on the presence or absence of asthma or atopy and re-analyzing the data. This will help determine if the variations in IFN production are more related to asthma, atopy, or both.

10. Table 1. The proportion of asthma patient with rhinitis was 12/22 but the percentage shown is 27%. Please clarify.

11. The methods state that RV infection was performed at MIO = 1. Please correct “MIO” to “MOI”.

12. The qPCR experiments are described as qPCR in some places and RT-PCR in other places. Important to be consistent. 

Comments on the Quality of English Language

Minor editing of English language required

Author Response

Reviewer 2

Summary

In this study, the authors recruited healthy and asthmatic adults from the human cohort study AZCRA. They found that the asthmatic subjects experienced up to six respiratory infections in the previous 12 months compared to a maximum of two infections for the healthy group. Nasal swabs were collected and cultured in the presence or absence of RV1B for 72 hours. They found that the cultures from asthmatic subjects had deficient IFNb and IFNL production especially after RV infection. The authors then tested the hypothesis that pretreatment with Poly-IC could restore IFN production and provide protection against infection. They found that pretreatment with Poly-IC 12 hours prior to RV infection enhanced TLR3 expression in A549 cells and antiviral immune responses. The authors conclude that poly I:C pre-treatment is a promising candidate therapeutic for the prevention of allergic asthma exacerbations.

Major comments

  1. Study design. The study design is described as a human cohort study however it was also registered as a clinical trial. The authors should clarify if this study is an observational study or a clinical trial. If the study is a clinical trial, they should provide details of the intervention.

Answer:

The AZCRA study is primarily an observational study whose objective is to compare healthy and asthmatics participants at a baseline (non-symptomatic) visit. It is also registered but it is not a clinical trial.

Reviewer

  1. Study population. The study population is described in Table 1. The table currently lacks some clinical variables that are important to describe the asthma patients. Some additional variables that could be included are:

Family history of atopy, Family history of asthma, Allergen-specific IgE levels, Total IgE levels, SPT to panel of aeroallergens, Lung function (spirometry), Current medications

Answer: We added the family history, the lung function measurements, and the asthma medications to the table. We don’t have the IgE levels nor aeroallergen data.

Reviewer

  1. Collection of nasal swabs. It is important to understand if the nasal swabs were collected when the patients were asymptomatic or not at the time of sampling. It is also important to understand if the patients were currently on medications at the time of sampling.

Answer:

The data shown in this paper is during our baseline visit, so while the patients were asymptomatic. Most of the asthmatic patients (77,26%) were on asthma medication. We included that information in our results.

Reviewer

  1. The authors measured the purity of the nasal epithelial cell cultures, but this was not discussed in the manuscript. The authors should present the purity data for asthmatic and healthy subjects before and after culturing and report on any differences between the groups.

Answer:

Due to limited cell counts of NEC, we only did the purity staining three times to show that it worked. Thus, we can’t report differences between the groups. In Fig. 1c,d displayed is the purity in control patients before and after culture.

Reviewer

  1. In figure 1e and 1f, IFN production in the RV stimulated cultures was lower than in the control cultures. This result was unexpected and is not consistent with the literature. The authors should check if the house keeping gene is stable and appropriate for this experimental system.

Answer: In the literature most cohort studies investigating RV responses use time points around 24h. In studies assessing more timepoints, the results showed that the IFN response peaks between 12 and 48h. However, due to protocol requirements we chose to investigate the 72h timepoint and wanted to see how the IFN levels develop at that timepoint. In future studies various timepoints could be included to investigate the time course of IFN levels. This difference in the experimental design might explain the inconsistency with the literature.
Regarding the house keeping gene HPRT: we use HPRT in all our AZCRA qPCR analysis to keep them comparable. In the literature, HPRT has already been used as a house keeping gene in primary airway epithelial cells (e.g. (4)).

Reviewer

  1. RV loads. The authors should measure RV loads using qPCR throughout the study to determine if this is higher in nasal epithelial cell cultures from subjects with asthma and additionally if the viral loads are altered by Poly-IC pretreatment.

Answer: We added the graph for the RV load in NEC to Figure 1. It was higher in asthmatic patients then in control patients. We also added the RV load to Figure 3, here we report slightly lower RV loads in the pretreatment condition.

Reviewer

  1. In the discussion, the authors should comment on the utility of Poly-IC in clinical settings and perhaps discuss other options such as inhaled IFNb or Poly-ICLC.

Answer: We added information about poly I:C and Poly-ICLC to the discussion.

Reviewer

  1. The authors conclude that poly I:C pre-treatment is a promising candidate therapeutic for the prevention of allergic asthma exacerbations. However, this conclusion is not supported by the data given that the atopic status of the asthma patients was not confirmed and only 12/22 of the asthmatic subjects have allergic rhinitis.

Answer: Thank you for pointing this it. We are sorry that we inserted the wrong data for allergic rhinitis, and we have now corrected it in the manuscript (see comment 10). 17 out of 22 patients had allergic rhinitis. Additionally, at the baseline visit we assessed the known asthma phenotype in our patients. 100% of our patients had either allergic asthma or mixed asthma, which included allergic asthma in combination with another phenotype (Table 1 and S1).

Reviewer

  1. The authors could consider stratifying the subjects into four groups based on the presence or absence of asthma or atopy and re-analyzing the data. This will help determine if the variations in IFN production are more related to asthma, atopy, or both.

Answer: We added the clinical data that all our asthma patients’ asthma phenotype is allergic or includes allergic asthma and we corrected that 77,26% of our patients have allergic rhinitis. In our questionnaire all but two asthma patients answered that they have clinically relevant allergies. In those two patients one has atopic eczema. In our control group we only recruited non-atopic patients. Thus, we unfortunately only have the two groups non-asthmatic, non-atopic and asthmatic, atopic.

  1. Table 1. The proportion of asthma patient with rhinitis was 12/22 but the percentage shown is 27%. Please clarify.

Answer: We are sorry for the error in our clinical data. The correct data is 17 out of 22 patients had allergic rhinitis, which is 77,26%.

  1. The methods state that RV infection was performed at MIO = 1. Please correct “MIO” to “MOI”.

Answer: Unfortunately, we can’t find this error in the manuscript.

Reviewer

  1. The qPCR experiments are described as qPCR in some places and RT-PCR in other places. Important to be consistent. 

Answer: The reviewer made a correct observation. We changed it consistently.

Reviewer

Comments on the Quality of English Language: Minor editing of English language required.

Answer

We improved this.

Reviewer 3 Report

Comments and Suggestions for Authors

Typographical error at table no 1 title, section 2.2 and 2.4 

Not giving abbreviation detail when introducing anything new such as AZURA, moi, etc. Quality of figure 1 can be improved. I was wondering if you have actual viral load on nasal swabs from RV positive patient because it would be interesting comparison to see viral load and their immune response to produce cytokines that can translate in vitro experiment.

What is the significance of Ep Cam cells? Authors should describe it.

Quality of figure 1 is not up to mark and that can be improved to understand better. I would like to ask author did you checked any other inflammatory cytokines such as IL-1beta, tnf-alpha or IL-6. Also did you check acutal RV virus load in those patients before assigning them. 

Comments on the Quality of English Language

Need modification

Author Response

Reviewer 3

Comments and Suggestions for Authors

Typographical error at table no 1 title, section 2.2 and 2.4 

Answer: In the title of table 1 we changed cohort to cohorts and added quotes to show that AZCRA is the name of the study. Also, we corrected sections 2.2 and 2.4 and hope that we removed the typographical errors the reviewer meant.

Reviewer

Not giving abbreviation detail when introducing anything new such as AZURA, moi, etc. Quality of figure 1 can be improved. I was wondering if you have actual viral load on nasal swabs from RV positive patient because it would be interesting comparison to see viral load and their immune response to produce cytokines that can translate in vitro experiment.

Answer: We have now added explanations to all abbreviations and included a paragraph about the AZCRA study and our goal to the introduction. We reworked figure one and hope you find it to be improved and clearer. data presented here was during our non-symptomatic baseline visit and it was conducted during Covid when everyone wore masks.

Reviewer

What is the significance of Ep Cam cells? Authors should describe it.

Answer: By showing the FACS data on EpCam expression on the cells, we wanted to show that the cells we cultured from our nasal swabs are mainly epithelial cells and that the cells are proliferating in culture. EpCam it is a multi-functional transmembrane protein that stands for epithelial cell adhesion molecule and is expressed in varying degrees on all epithelial cells. We did virus detection analysis in the nasal fluids of our patients_ however, none had any respiratory virus detectable. It can be used to analyze regulation of cell adhesion, proliferation, migration, stemness.We added this information to the manuscript as well.

Reviewer

Quality of figure 1 is not up to mark and that can be improved to understand better. I would like to ask author did you checked any other inflammatory cytokines such as IL-1beta, tnf-alpha or IL-6. Also did you check acutal RV virus load in those patients before assigning them.

Answer: We added abbreviations to the box plots and changed the arrangement, we hope it is easier to understand now.
In this part of our study, we did not check other inflammatory cytokines since we wanted to concentrate on interferon responses and that pathway.
As indicated above, we did not detect RV virus in the nasal fluids of our patients. Most patients did the baseline visit during Covid and during the time they had to wear masks.

The authors want to thank the three reviewers for their helpful and constructive comments that helped to improve our paper.

Literatur

  1. Levine AS, Levy HB. Phase I-II trials of poly IC stabilized with poly-L-lysine. Cancer Treat Rep. 1978;62(11):1907-12.
  2. De Waele J, Verhezen T, van der Heijden S, Berneman ZN, Peeters M, Lardon F, et al. A systematic review on poly(I:C) and poly-ICLC in glioblastoma: adjuvants coordinating the unlocking of immunotherapy. Journal of Experimental & Clinical Cancer Research. 2021;40(1):213.
  3. Saxena M, Sabado RL, La Mar M, Mohri H, Salazar AM, Dong H, et al. Poly-ICLC, a TLR3 Agonist, Induces Transient Innate Immune Responses in Patients With Treated HIV-Infection: A Randomized Double-Blinded Placebo Controlled Trial. Front Immunol. 2019;10:725.
  4. Zhu Y, Chew KY, Wu M, Karawita AC, McCallum G, Steele LE, et al. Ancestral SARS-CoV-2, but not Omicron, replicates less efficiently in primary pediatric nasal epithelial cells. PLoS Biol. 2022;20(8):e3001728.

Round 2

Reviewer 2 Report

Comments and Suggestions for Authors

The authors have addressed my comments and this has strengthened the manuscript. I have no further comments. 

Author Response

N/A